# Characterization of Microbial Communities and Naturally Occurring Radionuclides in Soilless Growth Media Amended with Different Concentrations of Biochar

**George K. Osei [1,\*], Michael Abazinge [1], Lucy Ngatia [2], Ashvini Chauhan [1], Alejandro Bolques [1], Charles Jagoe [1] and Ashish Pathak [1]**

[1]  School of the Environment, Florida A&M University, Tallahassee, FL 32307, USA
[2]  College of Agriculture and Food Sciences, Florida A&M University, Tallahassee, FL 32307, USA
[\*]  Correspondence: oseikofige@gmail.com

**Abstract:** Biochar, derived from the pyrolysis of plant materials has the potential to enhance plant growth in soilless media. Howevetar, little is known about the impact of biochar amendments to soilless growth media, microbial community composition, and fate of chemical constituents in the media. In this study, different concentrations of biochar were added to soilless media and microbial composition, and chemical constituents were analyzed using metagenomics and gamma spectroscopy techniques, respectively. Across treatments, carboxyl-C, phenolic-C, and aromatic-C were the main carbon sources that influenced microbial community composition. *Flavobacterium* (39.7%), was the predominantly bacteria genus, followed by *Acidibacter* (12.2%), *Terrimonas* (10.1%), *Cytophaga* (7.5%), *Ferruginibacter* (6.0%), *Lacunisphaera* (5.9%), *Cellvibrio* (5.8%), *Opitutus* (4.8%), *Mucilaginibacter* (4.0%) and *Bryobacter* (4.0%). Negative relationships were found between *Cytophaga* and $^{226}$Ra (r = −0.84, p = 0.0047), $^{40}$K (r = −0.82, p = 0.0069) and $^{137}$Cs (r = −0.93, p = 0.0002). Similarly, *Mucilaginibacter* was negatively correlated with $^{226}$Ra (r = −0.83, p = 0.0054) and $^{137}$Cs (r = −0.87, p = 0.0021). Overall, the data suggest that high % biochar amended samples have high radioactivity concentration levels. Some microorganisms have less presence in high radioactivity concentration levels.

**Keywords:** biochar; radioactivity; thermal stability; pyrolysis; soilless

## 1. Introduction

The use of soilless substrate as plant growth media over the past five and half decades has attracted high attention globally [1]. Organic (peat, compost, coir, bark, and wood fiber) and inorganic (Rockwool volcanic rock, tuff, expanded clay granules, vermiculite, zeolite, and pumice) materials have been utilized as soilless substrates, individually and in combination, and with additives such as fertilizers [2]. In the past, peat has been the major material used in container agriculture. But recent concerns about its economic cost and environmental implications [3], have led to renewable organic materials being used as substitutes. Some common renewable materials currently used as soilless substrates documented in the scientific literature are coconut (*Cocos nucifera* L.) husk fiber (coir), ground pine (*Pinus taeda* L.) bark and logs, rice (*Oryza sativa* L.) hulls, and switchgrass (*Panicum virgatum* L.) [1]. The physical and chemical properties of substrates can be affected by their inherent components [4], and these physicochemical properties can be manipulated with the use of additives such as biochar.

Biochar is a carbon (C) rich charred organic product derived from the pyrolysis of waste biomass in the absence of, or little oxygen [5]. Extensive studies and reviews have been conducted on biochar amendment to traditional or mineral soil [6]. The type of biomass, pyrolysis temperature, and storage conditions influence the physicochemical properties of biochar [7]. Biochar influences the physicochemical properties of soil by increasing soil pH to improve soil fertility, changing soil bulk density, and improving organic C and cation

exchange capacity [8]. Additional benefits of biochar are soil water retention, improved nutrient retention to increase crop yield, and altered microbial populations and functions in soil [5]. The resistance of biochar to microbial degradation has been highlighted by [9]; this limits the release of C in the form of $CO_2$ into the atmosphere to mitigate climate change.

Various environmental factors such as moisture content, temperature, and organic matter determine the abundance and activities of microorganisms in soil [10]. Soil microbial communities play a vital role in nutrient acquisition [11]. Microbial communities are considered important biological indicators of terrestrial ecosystem stress because of their sensitivity to abiotic changes, soil quality, and plant cover [10]. Understanding the composition and diversity of microbial communities in relation to environmental parameters is very important [12] and soil microbial community structure and activity, can be altered and enhanced to improve soil properties by biochar amendment [13].

Radionuclides can either be natural or anthropogenic [14]. The three major primordial radionuclides are $^{238}$U and $^{232}$Th, and $^{40}$K, and these occur naturally in minute concentrations depending on the geological, and geographical nature of the parent rock and soil [14,15]. Atmospheric nuclear weapons testing, nuclear accidents such as Chernobyl and Fukushima, and mining activities have been the primary sources of global anthropogenic radioactive contaminants in the environment for the past seven and half decades [16–18].

The release of radioactive materials into the environment leading to exposure to the population has stimulated intense public concern and has substantially led to research on the fate of major radionuclides in the environment [19,20]. Radionuclides in growth media for agriculture and horticulture crop production can potentially be a long-term source of radiation exposure to humans and the environment due to their accumulation by plants [21].

Literature has shown that the mobility and interaction of a range of radionuclides in both natural and engineered environments are influenced by microorganisms [20]. Microbial communities, since the origin of cellular life, have periodically been exposed to contaminated environments [22]. While there have been numerous studies of microbe-radionuclide interactions in agricultural soils [15,23–25], to our knowledge, there have not been any studies in biochar-amended soilless growth media on the relationships among microbial community diversity and the activity of radionuclides.

With fixed land area and a growing population worldwide, the causes and effects of the deterioration of agricultural land have been debated elsewhere [26]. Globally, anthropogenic activities are converting natural land cover into human-dominated ecosystems [27]. It has been reported that approximately 13 million hectares of natural land cover were converted to other land uses each year between 2000 and 2010 [27,28]. However, as a potential mitigation strategy for the deterioration of agricultural lands, soilless-biochar amendment systems for agriculture are gaining increasing attention globally [29,30].

However, studies involving relationships among C composition, microbial communities, and radionuclides in soilless growth media amended with biochar are in their infancy. For this reason, we examined C composition, bacteria composition, and radionuclide activities in soilless growth media amended with biochar samples collected from Florida Agriculture and Mechanical University Research and Extension Center (FAMU-REC) located in Quincy, Florida (FL). The aim of the present study was to determine the relationships among (i) C composition and thermal stability (R400) and microbial diversity, and (ii) microbial diversity and radionuclides activity levels, in growth media amended with biochar samples. The present study is the first to provide baseline data among these parameters in biochar-amended growth media, not only for FAMU-REC, but for Florida, the country, and the world.

## 2. Materials and Methods

### 2.1. Samples Collection Site Description

The study site where samples were collected was the FAMU-REC in Quincy, Gadsden County, FL in the USA. The site on the Florida-Georgia State line (30°67′ N and 84°61′ W)

is approximately 30 miles from the main university campus in Tallahassee, FL. The FAMU-REC consists of more than 100 acres of farmland, pines forest, lakes and animal research facilities, and laboratories. It has annual high and low average temperatures of 26.1 °C (79.0° F) and 12.9 °C (55.3° F), respectively, with an annual average temperature of about 19.5 °C (67.2° F), annual precipitation of 59.67 inches, and humidity level of approximately 94.0%.

### 2.2. Media Composition

A soilless growth media consisting of a mixture of 60% coconut coir and 40% fine pine bark was used to prepare 8 biochar treatments containing 1%, 2%, 3%, 4%, 6%, 8%, 10%, and 12% biochar, respectively, plus a control, without biochar. Compressed coconut coir bricks of 8 × 4 × 2-inches were expanded and rehydrated by soaking in water, expanding volume to 5–7 times the original size. A cement mixer was used to mix the soilless media treatment and biochar amendments, which were dispensed into 3-gallon plastic containers to give 3 replicates per treatment. Triplicates of each treatment were collected for analysis and the means were reported.

### 2.3. Sample Analysis

2.3.1. pH Determination

1 g of sample was placed in 20 mL of deionized water (DI), shaken for 1.5 h, and then left for 5 min to equilibrate before measuring pH with a Fisher Scientific Accumet Basic AB15 pH meter [31–33].

2.3.2. Nuclear Magnetic Resonance Analysis (NMR)

Nuclear magnetic resonance technique was used to evaluate the carbon composition of the soilless media's carbon functional groups. Finely milled powder samples of soilless media amended with various percentages of biochar by weight were analyzed using the magic angle spinning $^{13}$C solid-state nuclear magnetic resonance (MAS $^{13}$C $_{SS}$NMR) technique, as previously employed by [31,34] using a Bruker 300 MHz DR NMR spectrometer equipped with a Bruker 4.0 mm double resonance NMR probe.

2.3.3. Multi-Elemental Scanning Thermal Analysis (MESTA)

Sample total C content was determined using the multi-elemental scanning thermal analysis (MESTA) technique previously employed by [31,35,36]. Carbon thermograms were created using C content. Due to the high C concentration in biochar [31], a dilution consisting of a 1:5 mixture of sample: talc by weight was applied before the MESTA analysis to improve thermogram resolution. The analyses were performed using Antek 9000HN Series Nitrogen Analyzer by SpectraLab Scientific Inc., Markham, ON Canada.

Low and high carbon stability were examined for C recovered at temperatures of <400 °C and >400 °C, respectively [31]. R400 is the region identified below 400 °C normalization divided by the total surface area [37]. Using Equation (1) below, R400 was computed as the fraction of low thermal stability to total C of samples based on the data acquired from the low C thermal stability (<400 °C):

$$\text{``R400''} = \frac{(C \; recovered \; at < 400 \; degree)}{TC} \tag{1}$$

2.3.4. DNA Extraction, Quantification, and Purity, Metagenomics

Samples' genomic DNA was extracted using DNeasy Powersoil Kit (QIAGEN Inc., Germantown, MD, USA) per the manufacturer's instruction. A NanoDrop 1000 (NanoDrop Technologies, Wilmington, DE, USA) was used to quantify the total DNA of the extracted samples by measuring the concentrations (ng/µL) by absorbance at A260/280, and A260/230 ratios [38]. Sequence libraries were prepared using universal primers 345F (GTGCCAGCMGCCGCGGTAA) and 371R (CCGYCAATTYMTTTRAGTTT) to perform the

amplification of the16S r RNA metagenomics. A mid-output kit with $2 \times 150$ paired-end sequencing was utilized using Illumina MiSeq437 equipment to do the sequencing [39].

PEAR was used to integrate forward and backward reads [40]. Based on a quality threshold of $p = 0.01$, combined readings were edited to remove ambiguous nucleotides and primer sequences. Any reads with sequences shorter than 300 bp or without a primer sequence were eliminated. Using the USEARCH algorithm and a comparison to the Silva v132 reference sequence database, chimeric sequences were found and eliminated [41,42].

To produce taxonomy summaries utilizing a sub-OTU resolution of the sequence collection, the conventional QIIME pipeline was modified [43,44]. The generated sequence files were then quickly combined with the sample data. The list of unique sequences was then produced by dereplicating each sequence. All sequences with at least 10 counts of abundance were referred to as seed sequences. The next step was to use USEARCH to locate the closest seed sequence for any non-seed sequence that met the 97% minimum identity requirement. If a non-seed sequence matched a seed sequence, its counts were combined with the counts from the seed sequence, and if it didn't, it remained an independent sequence [41].

With a minimum similarity threshold of 90%, taxonomic annotations for seed and mismatched non-seed sequences were assigned using the USEARCH and Silva v132 reference [41,42]. The usual QIIME assignment algorithm was changed to only consider hits at each taxonomic level with an assigned name to increase the depth of annotation. When assigning the taxonomic kingdom through the family, a reference annotated as "k Bacteria; p Firmicutes; c Clostridia; o Clostridiales; f Ruminococcaceae; g; s_" would be considered, but it would not be used when assigning the genus or species. Additionally, to be considered for genus or species level designation, any hits in the reference database must have a minimum identity of 97% or 99%, respectively. Then, sequence abundance data and taxonomic annotations were combined into a single sequencing table.

### 2.3.5. Radionuclides Sample Preparation and Analysis

Samples of the soilless media with the various biochar amendments were dried at 110 °C for 48 h in an electric oven to remove moisture content [45]. The dried samples were transferred into 500 mL Marinelli beakers of the same geometry as the reference material, covered, sealed with parafilm to limit any possible escape of radon, and relabeled as above. The prepared samples were left for at least 30 days to reach secular equilibrium with radon and its daughters [45–47]. Each sample was handled carefully, and proper measures were taken to prevent cross-contamination.

Gamma spectrometry analyses were performed using high purity germanium (HPGe); detector by Canberra Industries/Merion Inc., Meriden, CT. The detector was shielded with a thick lead shield with Cu inner layer to reduce the detection of background radiation. The HPGe detector was coupled with a Canberra DSA-2000 data acquisition system and connected directly with a PC equipped with Canberra Genie 2000 software in which measured gamma spectra were stored and analyzed. The software internally calculates activity concentrations of radionuclides from all prominent gamma lines with background subtraction [48]. The instrument has an energy resolution of 0.5keV full width at half of the maximum (FWHM) for a 1332 keV channel (using Co-60) and a relative photopeak efficiency of 35%. The instrument was calibrated for energy and efficiency over the photon energy range of 2 to 2000 keV using a National Institute of Standards and Technology (NIST) traceable mixed gamma standard. Each sample was counted for a period of 86,400 s.

### 2.3.6. $^{235}$U, $^{226}$Ra, $^{232}$Th, $^{40}$K and $^{137}$Cs

The activity concentrations of the natural radionuclides $^{235}$U and $^{40}$K, and the anthropogenic radionuclide $^{137}$Cs were determined directly from their photopeak energies lines of 185.7 (54.0%) 1460.8 (10.7%), and 661.7 (85.1%) keV, respectively [45]. The weighted mean photopeak energy lines of $^{214}$Pb (295.2 and 351.9) and $^{214}$Bi (609.3 and 1120.3 keV) were used to estimate the activity concentration value of $^{226}$Ra. The weighted mean pho-

topeak energies lines of [212]Pb (238.6 keV), [212]Bi (727.2 keV), and [228]Ac (338.3, 911.6, and 969.1 keV) were used to determine the activity concentration value of [232]Th [46,49]. Using the weighted mean photopeak procedure for multiple energy lines gives more accurate results with lower errors than using only one of the photopeak lines [50]. The measured and estimated activity concentration values of the radionuclides are reported in $Bqkg^{-1}$.

### 2.3.7. Metagenomic Sequence Accession Numbers

The 16S metagenomic sequences obtained from this study are available from NCBI's Sequence Read Archive under Bioproject accession # PRJNA773140.

### 2.3.8. Statistical Analysis

JMP software (version 13.2.1, SAS, Cary, NC 27513, USA) was used to conduct ANOVA and Pearson correlation analysis for this study. Data are reported as means and standard error of the mean (SEM). Analysis of variance using post-hoc Turkey HSD tests, where $\alpha = 0.05$, was used to determine significant differences among treatment variables. Sigma plot (version 12.0) was used to plot correlations between variables.

## 3. Results

### 3.1. Samples' Basic Physicochemical Characteristics

Physicochemical characteristics of growth media samples from control (0% biochar) and different percentages of biochar amendments are reported in (Table 1). pH increased with an increasing percentage of biochar amendments, with values ranging between 6.02–6.84. (Table 1). Similarly, TC, C composition; carboxyl-C, phenolic-C, and aromatic-C, increased with the increasing percentage of biochar amendments to growth media. TC, carboxyl-C, phenolic-C, and aromatic-C, concentration values ranged between 443.03 to 514.47, 14.58 to 38.29, 32.17 to 56.11, and 65.66–95.59 g $kg^{-1}$, respectively (Table 1).

**Table 1.** Samples' physicochemical and C composition properties with TC, carboxyl-C, phenolic-C, and aromatic-C in g $kg^{-1}$.

| Biochar | pH | TC | Aromatic | Phenolic | Carboxyl | R400 |
|---------|------|--------|----------|----------|----------|------|
| 0% | 6.15 | 443.67 | 67.76 | 32.17 | 14.89 | 0.61 |
| 1% | 6.02 | 447.80 | 67.96 | 39.82 | 17.48 | 0.70 |
| 2% | 6.03 | 443.03 | 65.66 | 34.72 | 14.58 | 0.65 |
| 3% | 6.23 | 454.60 | 73.98 | 42.32 | 27.72 | 0.67 |
| 4% | 6.47 | 470.17 | 78.04 | 42.40 | 23.06 | 0.65 |
| 6% | 6.46 | 459.97 | 73.88 | 42.24 | 20.59 | 0.62 |
| 8% | 6.68 | 478.40 | 82.16 | 48.34 | 25.40 | 0.60 |
| 10% | 6.76 | 492.43 | 92.60 | 53.34 | 26.38 | 0.59 |
| 12% | 6.84 | 514.47 | 95.59 | 56.11 | 38.29 | 0.50 |

TC = total carbon.

The R400 value for the growth media without added biochar was 0.61. with added biochar, the R400 values ranged from 0.50 to 0.70. However, the 1% biochar amended growth media recorded the highest R400 value: 0.7. The values then tend to decrease with an increasing percentage of biochar amendments (Table 1). Generally, R400 exhibited a declining trend with increasing biochar percentage amendments, decreasing from 0.7 for 1% biochar to 0.5 for 12% biochar (Table 1).

Cluster analysis revealed close similarity among M03, M04, M12, M08, and M10, representing 3%, 4%, 12%, 8%, and 10% biochar amendments, respectively (Figure 1). M0, M1, M2, and M6, representing biochar amendments of 0, 1, 2, and 6%, respectively, clustered independently. These findings indicate that biochar modifications had an impact on media qualities.

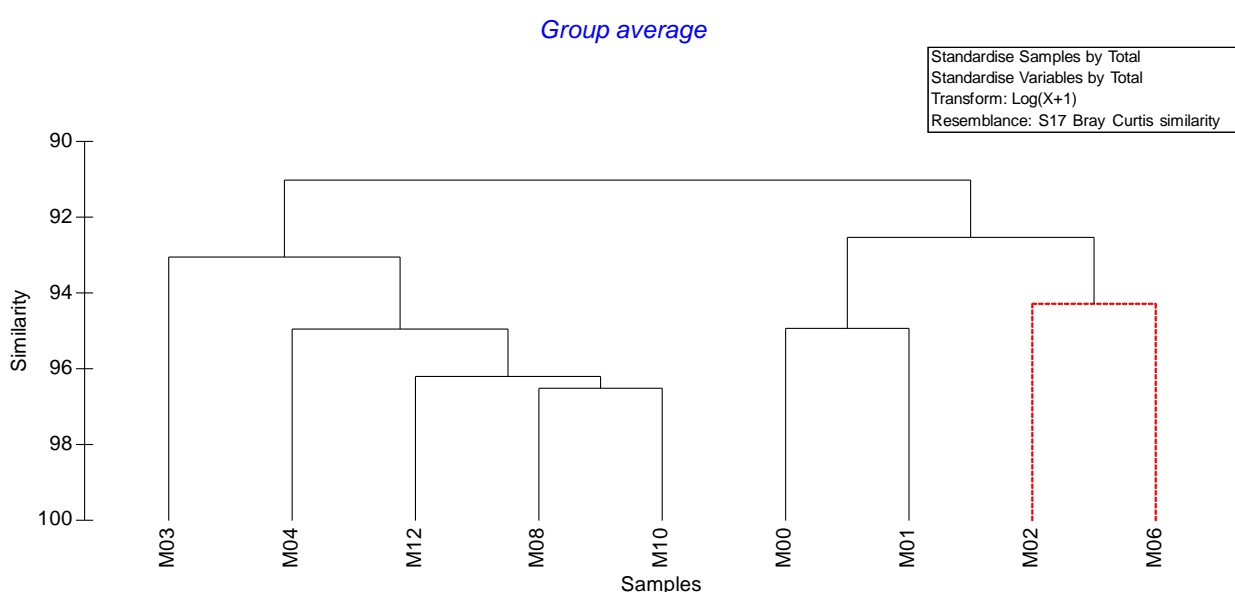

**Figure 1.** Cluster analysis without considering biochar media amendments as a variable.

### 3.2. Media Microbial Composition

Sample microbial composition and diversity were evaluated by 16S rRNA gene sequencing-based metagenomics. *Proteobacteria* (41.6%), *Bacteroidetes* (27.0%), *Verrucomicrobia* (7.9%), *Acidobacteria* (5.9%), *Planctomycetes* (3.9%), *Actinobacteria* (2.9%), *Chloroflexi* (2.2%), *Cyanobacteria* (1.4%) and *Patescibacteria* (1.3%) were the 9 predominant bacteria phyla in all samples. The phyla level bacteria relative abundance is reported in Figure S1. At the genera level, *Flavobacterium* (39.7%) was the dominant bacteria identified in the samples, followed by *Acidibacter* (12.2%), *Terrimonas* (10.1%), *Cytophaga* (7.5%), *Ferruginibacter* (6.0%), *Lacunisphaera* (5.9%), *Cellvibrio* (5.8%), *Opitutus* (4.8%), *Mucilaginibacter* (4.0%) and *Bryobacter* (4.0%), with their relative abundance shown in Figure 2.

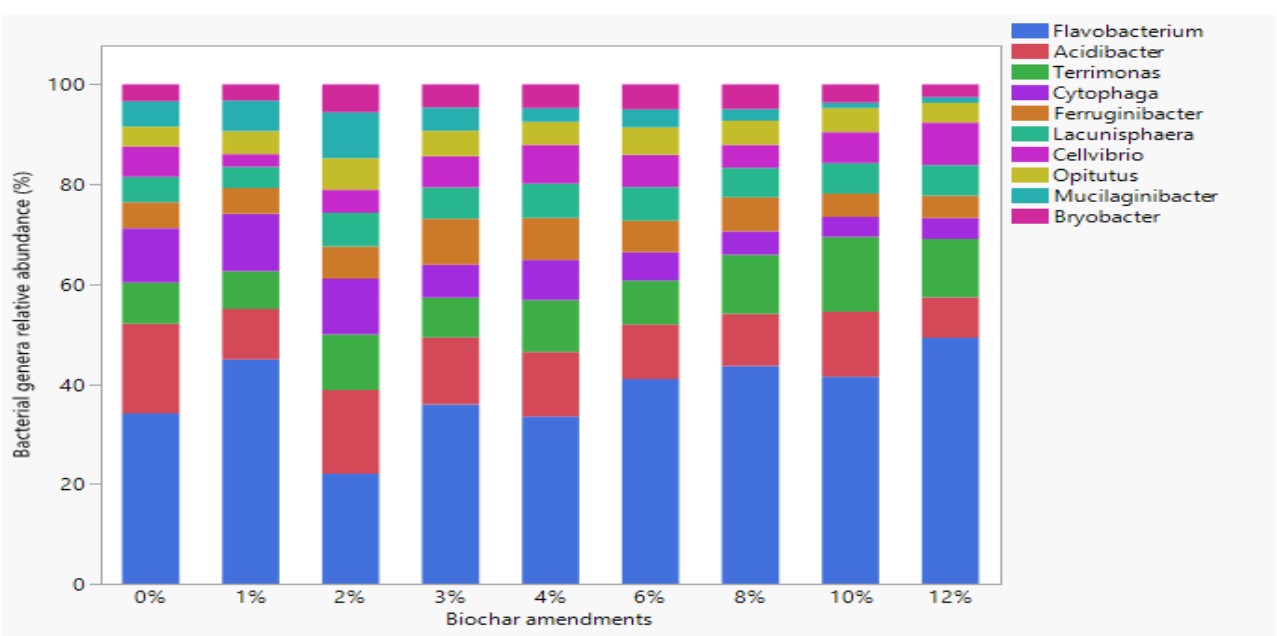

**Figure 2.** Relative abundance of dominant bacteria genera in different % biochar amended soilless growth media.

### 3.3. Radioactivity Measurements

Radionuclide activities are summarized in Table 2. $^{235}U$, $^{226}Ra$, $^{232}Th$, and $^{40}K$ were detected as natural radionuclides, as was anthropogenic $^{137}Cs$. The natural radionuclide with the highest recorded activity concentration (Bq kg$^{-1}$) was $^{40}K$, followed by $^{226}Ra$, $^{232}Th$, and $^{235}U$ in that order. Interestingly, even though $^{40}K$ recorded the highest activity concentration in the analyzed samples, its presence was not detected in the control and 1% biochar amended samples. The activity concentrations for $^{235}U$, $^{226}Ra$, $^{232}Th$, $^{40}K$ and $^{137}Cs$ ranged between 0.81–0.99, 2.54–4.14, 0.57–1.05, 19.57–24.23 and 0.19–0.55 Bq kg$^{-1}$, respectively (Table 2). Activity concentrations of $^{226}Ra$ and $^{137}Cs$ increased with an increasing percentage of biochar amendment to control media. Correlation analysis of radionuclides contents indicated that $^{226}Ra$ and $^{40}K$ concentrations correlated significantly (r = 0.67, p = 0.0497), suggesting $^{226}Ra$ and $^{40}K$ have dissolution similarities in media (Table 3). Similarly, $^{226}Ra$ and $^{137}Cs$ (r = 0.83, p = 0.0052), $^{40}K$ and $^{137}Cs$ (r = 0.74, p = 0.0231) concentrations significantly correlated (Table 3). However, $^{235}U$ and $^{232}Th$ were not significantly correlated with the other radionuclides (Table 3).

**Table 2.** Activity Concentration of Radionuclides (Bq kg$^{-1}$) from FAMU-RCE growth media.

| Biochar | $^{235}U$ | $^{226}Ra$ | $^{232}Th$ | $^{40}K$ | $^{137}Cs$ |
|---------|-----------|------------|------------|----------|------------|
| 0% | 0.92 ± 0.05 | 2.93 ± 0.12 | 1.01 ± 0.04 | 0.00 | 0.21 ± 0.02 |
| 1% | 0.92 ± 0.05 | 2.54 ± 0.11 | 0.68 ± 0.08 | 0.00 | 0.19 ± 0.02 |
| 2% | 0.99 ± 0.05 | 2.94 ± 0.12 | 1.02 ± 0.11 | 22.54 ± 0.54 | 0.22 ± 0.02 |
| 3% | 0.94 ± 0.05 | 3.10 ± 0.13 | 0.98 ± 0.18 | 21.10 ± 0.52 | 0.39 ± 0.03 |
| 4% | 0.92 ± 0.05 | 3.78 ± 0.14 | 0.57 ± 0.09 | 21.37 ± 0.53 | 0.37 ± 0.03 |
| 6% | 0.81 ± 0.04 | 2.89 ± 0.13 | 0.90 ± 0.04 | 19.57 ± 0.51 | 0.39 ± 0.03 |
| 8% | 0.90 ± 0.05 | 3.63 ± 0.14 | 0.88 ± 0.10 | 22.18 ± 0.54 | 0.48 ± 0.03 |
| 10% | 0.96 ± 0.05 | 3.93 ± 0.14 | 1.05 ± 0.10 | 24.23 ± 0.57 | 0.45 ± 0.02 |
| 12% | 0.94 ± 0.10 | 4.14 ± 0.29 | 1.02 ± 0.22 | 23.70 ± 1.13 | 0.55 ± 0.06 |

**Table 3.** Correlation matrix of detected radionuclides in analyzed samples.

| | $^{235}U$ | $^{226}Ra$ | $^{232}Th$ | $^{40}K$ | $^{137}Cs$ |
|---|-----------|------------|------------|----------|------------|
| $^{235}U$ | 1.00 | | 0.16 | 0.13 | −0.15 |
| $^{226}Ra$ | 0.23 | 1.00 | −0.13 | 0.67 | 0.83 |
| $^{232}Th$ | 0.16 | −0.13 | 1.00 | 0.17 | 0.11 |
| $^{40}K$ | 0.13 | 0.67 | 0.17 | 1.00 | 0.74 |
| $^{137}Cs$ | −0.15 | 0.83 | 0.11 | 0.74 | 1.00 |

### 3.4. Media Physicochemical Properties Relationship with Bacterial Composition and Radionuclides Activity Concentrations

Figure 3 shows that pH was related both positively and negatively to the percentage of various bacteria phyla in treatments. Specifically, media pH positively correlated with *Chloroflexi* (r = 0.73, p = 0.0268) and negatively correlated with *Cyanobacteria* (r = −0.70, p = 0.0368) (Figure 3a,b, respectively). At the bacterial genera level, there were significant negative correlations between pH and the genera *Cytophaga* (r = −0.91, p = 0.0007) and *Mucilaginibacter* (r = −0.90, p = 0.0008) (Figure 3c,d, respectively). Similarly, Figure 3e,f, respectively, show that pH was positively correlated with radionuclides $^{226}Ra$ (r = 0.89, p = 0.0012) and $^{137}Cs$ (r = 0.93, p = 0.0002).

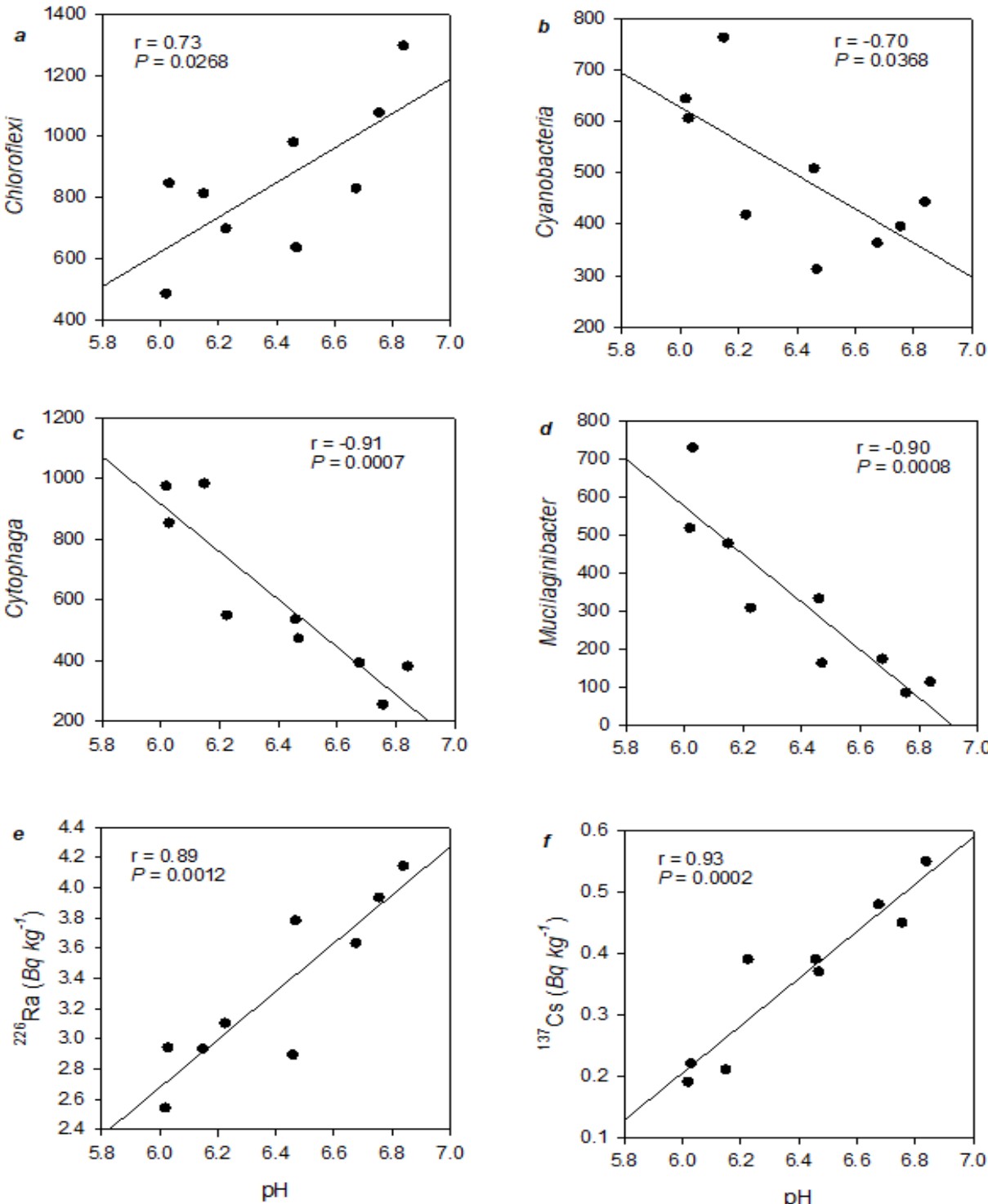

**Figure 3.** Correlations of pH with number of genotypes detected in various bacterial phyla: (**a**) *Chloroflexi* and (**b**) *Cyanobacteria*, genera: (**c**) *Cytophaga* and (**d**) *Mucilaginibacter*, and correlations of pH with radionuclide activity concentrations: (**e**) $^{226}$Ra and (**f**) $^{137}$Cs.

Additionally, C composition showed relationships with bacterial composition. As reported in Figure S2, TC, phenolic-C, and aromatic-C and bacterial phyla *Acidobacteria*, *Chloroflexi* and *Cyanobacteria* exhibited positive and negative relationships. *Acidobacteria* exhibited a significant negative relationship with aromatic-C (r = −0.67, *p* = 0.0461) and phenolic-C (r = −0.68, *p* = 0.0436) (Figure S2a,b, respectively). In contrast to *Acidobacteria*, the bacteria phylum *Chloroflexi* was positively correlated with aromatic-C (r = 0.73, *p* = 0.0261), and TC (r = 0.94, *p* = 0.0225) (Figure S2c,d, respectively). Like *Acidobacteria*, there was a significant negative relationship between *Cyanobacteria* and phenolic-C (r = −0.70, *p* = 0.0369) (Figure S2e).

Furthermore, the genus *Cytophaga* was negatively correlated with carboxyl-C (r = −0.77, *p* = 0.0154) (Figure S3a), phenolic-C (r = −0.87, *p* = 0.0026) (Figure S3b) and aromatic-C (r = −0.86, *p* = 0.0027) (Figure S3c). Similarly, the genus *Mucilaginibacter* was negatively correlated with carboxyl-C (r = −0.79, *p* = 0.0114), phenolic-C (r = −0.84, *p* = 0.0045), and aromatic-C (r = −0.87, *p* = 0.0021) (Figure S3d–f, respectively). Aromatic-C and phenolic-C were correlated with the bacteria genera *Terrimonas* (r = 0.70, *p* = 0.0375) and *Acidibacter* (r = −0.71, *p* = 0.0318), respectively (Figure S4a,b).

Significant relationships among C composition and radionuclides were positive correlated between carboxyl-C and $^{226}$Ra (r = 0.78, *p* = 0.0129), phenolic-C and $^{226}$Ra (r = 0.81, *p* = 0.0077), aromatic-C and $^{226}$Ra (r = 0.91, *p* = 0.0006) (Figure S5a–c). Carboxyl-C and $^{137}$Cs (r = 0.90, *p* = 0.0009), phenolic-C and $^{137}$Cs (r = 0.90, *p* = 0.0009) and aromatic-C and $^{137}$Cs (r = 0.89, *p* = 0.0011) again correlated positively (Figure S5d–f). Similarly, TC was positively correlated with $^{226}$Ra (r = 0.91, *p* = 0.0008) and $^{137}$Cs (r = 0.89, *p* = 0.0012) (Figure S6a,b). The results from this study indicated that higher values of C composition (carboxyl-C, phenolic-C, and aromatic-C) are associated with higher activity concentration values of $^{226}$Ra and $^{137}$Cs, and vice versa (Tables 1 and 2).

Relating to C thermal stability, which was translated to R400, bacterial phyla *Chloroflexi* (r = −0.92, *p* = 0.0004) and *Patescibacteria* (r = −0.71, *p* = 0.0335) both correlated negatively with R400, respectively (Figure S7a,b). Additionally, R400 was negatively correlated with the genera *Terrimonas* (r = −0.79, *p* = 0.0107) and *Cellvibrio* (r = −0.79, *p* = 0.0108) (Figure S7c,d). Similarly, R400 showed significant negative relationship with $^{226}$Ra (r = −0.74, *p* = 0.0227) and $^{137}$Cs (r = −0.72, *p* = 0.0292) (Figure S7e,f).

### 3.5. Samples Bacteria Composition Relationship with Radionuclides Contents

Bacterial phylum *Cyanobacteria* was negatively correlated with $^{226}$Ra (r = −0.73, *p* = 0.0270) (Figure 4a), $^{137}$Cs (r = −0.79, *p* = 0.0121) (Figure 4b) and $^{40}$K (r = −0.80, *p* = 0.0100) (Figure 4c). In contrast, *Chloroflexi* was positively correlated with $^{137}$Cs (r = 0.67, *p* = 0.0468) (Figure 4d). At the bacteria genera level, both *Cytophaga* and *Mucilaginibacter* negatively correlated with radionuclide activity concentrations. *Cytophaga was positively* correlated with $^{226}$Ra (r = −0.84, *p* = 0.0047), $^{40}$K (r = −0.82, *p* = 0.0069) and $^{137}$Cs (r = −0.93, *p* = 0.0002) (Figure 5a–c). Similarly, *Mucilaginibacter* was positively correlated only with $^{137}$Cs (r = 0.87, *p* = 0.0021) and $^{226}$Ra (r = −0.83, *p* = 0.0054) (Figure 5d,e). The negative correlations of the above bacteria phyla and genera with the radionuclides suggested that greater counts of these bacteria phyla and genera are associated with lower radionuclide content in the samples. However, there were no significant correlations for sample physicochemical parameters, and or for bacterial proportions with activity concentrations of $^{235}$U and $^{232}$Th.

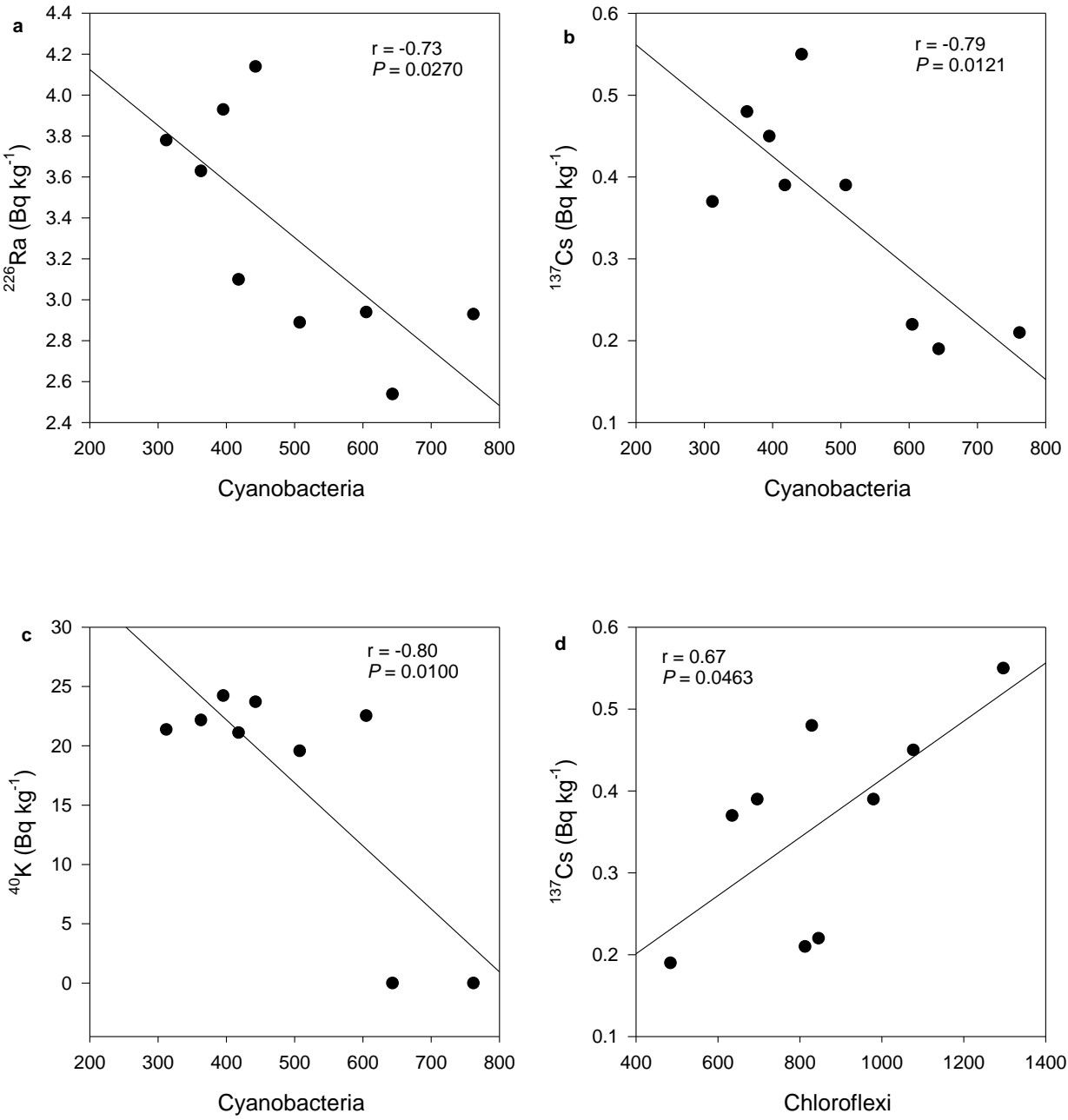

**Figure 4.** Relationships between number of bacterial genotypes detected from various phyla and activity concentrations of radionuclides. *Cyanobacteria* correlations with (**a**) [226]Ra, (**b**) [40]K and (**c**) [137]Cs. (**d**) *Chloroflexi* correlation with [137]Cs.

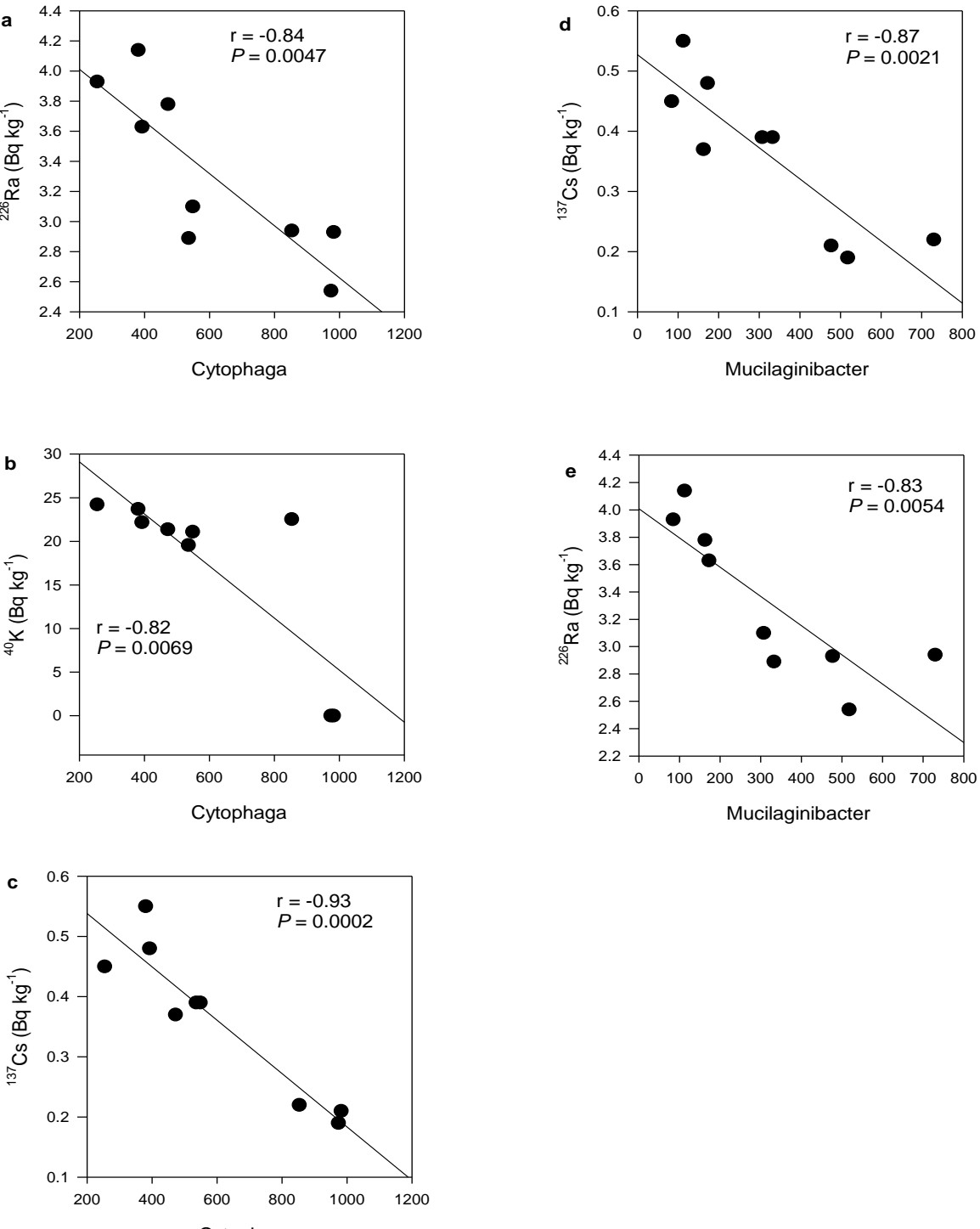

**Figure 5.** Correlations between number of genotypes detected from the genus *Cytophaga* with activity concentrations of (**a**) $^{226}$Ra, (**b**) $^{40}$K, and (**c**) $^{137}$Cs, and from the genus *Mucilaginibacter* with (**d**) $^{137}$Cs and (**e**) $^{226}$Ra.

## 4. Discussion

The physiochemical factor pH strongly influences bacterial diversity in soils [51,52]. Microbial populations can be strongly influenced by media pH some bacteria can adapt to extreme pH, although many enzymes and proteins can become denatured and inactivated under high acidic (<4.0) and alkaline (>9.0) conditions, impairing many metabolic processes [51]. Ref. [51] reported that most bacteria favor circumneutral pH conditions

(between 6.5 and 7.5). Amending biochar to media changed the media pH which, in turn, likely affected the bacterial community composition [53].

Despite the complex nature of environmental microbial communities, our results show patterns characterizing microbial abundance of Proteobacteria phylum in the biochar amended growth media. Literature has indicated the consistent abundance of *Proteobacteria* in many studied areas, and the relative abundance of *Proteobacteria*, regardless of the % biochar added to the growth media, was high compared to other bacterial phyla. *Proteobacteria* have been recognized as plant growth-promoting bacteria and facilitate nutrient acquisition and disease protection [54]. For that reason, a low relative abundance of *Proteobacteria* [54], can have a major effect on the plant productivity and media health in an agricultural environment. Proteobacteria are globally recognized for their important role in carbon cycling [55], consistent with the high abundance of *Proteobacteria* because of the high carbon concentration of biochar.

In agricultural systems, *Bacteroidetes* are abundant and noted for their ability to metabolize organic matter exploitation [56]. In the present study, the high abundance of *Bacteroidetes*, which was second only to *Proteobacteria*, regardless of the percentage of biochar added to growth media, agreed with previous findings [56]. According to [57,58], *Bacteroidetes* promote plant growth and plants' resistance to environmental stress. Other authors reported that *Bacteroidetes* are enriched in environments with high C availability [59], and hence their high abundance in biochar amendment media in the present study was not a coincidence. *Verrucomicrobia* was third in abundance in all the samples analyzed, ranked behind *Proteobacteria* and *Bacteroidetes* Based on their lower growth rates and adaptations for growing on relatively recalcitrant forms of C, a major component of biochar, and a major constituent of the growth media in the present study, *Verrucomicrobia* is classified as an "oligotroph" [60].

*Acidobacteria* is also another diverse group of bacteria found in soil [61,62] and has been affiliated with the biogeochemical cycling of C, a major constituent of biochar [63]. Additionally, *Acidobacteria* has been associated with the degradation of recalcitrant polymer and is considered an important phylum ecologically in the turnover of soil organic matter [61,64]. Therefore, its presence in the biochar-amended media in this study can be attributed to its association with recalcitrant C compounds (the main constituent of biochar) [63]. Similarly, previous research suggested that *Acidobacteria* prefer an acidic environment [65]. This was consistent with the high abundance of *Acidobacteria* in lower pH samples in this study. Furthermore, *Acidobacteria* exhibits an oligotrophic lifestyle [66] consistent with our study because of their negative correlation with carbon composition as previously suggested [66].

In an agricultural setting, *Actinobacteria* enhance plant growth and improve plant nutrition. They provide direct plant growth-promoting mechanisms such as nitrogen fixation to improve plant nutrition [67]. *Actinobacteria* are found widely in different environments and play a critical role in organic matter decomposition [55,68]. This is in line with its negative relationship with R400 observed in this study. Also, *Actinobacteria* are known for the production of antibiotics and other secondary metabolites to suppress other bacteria [69].

An essential macronutrient limiting agricultural productivity, N, is the largest and most costly input in agriculture [70]. Atmospheric and dissolved dinitrogen ($N_2$) are abundant in soil and water, yet unable for used by plants [70]. However, there is conclusive evidence that N is made available for plants and other organisms through N fixation by *Cyanobacteria* [71] Hence the presence of *Cyanobacteria* in this study was an important benefit in the biochar amended growth media for use in agriculture. In this study, *Chloroflexi* increased with increasing media parameters such as pH, carboxyl-C, phenolic-C, and aromatic-C, while *Cyanobacteria* decreased with increases in the same media parameters. This scenario agrees with an observation by [60] in agricultural systems of arid, continental, and temperate regions. It was reported that *Chloroflexi* metabolic flexibility can be a disadvantage in competition with *Cyanobacteria* for nutrient and physical space when they co-exist in the same environment [72].

*Flavobacterium*, the predominant bacterial genera found in samples collected from this study, has been shown to be widely distributed in nature and functions in different types of organic matter mineralization and rapid digestion of insoluble chitin [73,74]. The genus *Cellvibrio* has previously been reported to produce hydrolytic enzymes [73]. Media biochar amendment has been shown to stimulate the genera *Flavobacterium* and *Cellvibrio*. The genus *Cellvibrio* has been reported to promote plant growth and act as an inducer of plant systemic resistance [73]. This is in line with what resulted in this study where *Cellvibrio* was negatively correlated with R400 ($p = 0.0108$), translating into the decomposition of organic matter for available use in crops. The genus *Cytophaga* belongs to the phylum *Bacteroidetes* and is known for its ability to degrade cellulose [75] and other high molecular weight organic compounds [76].

In this study, increasing the percentage of biochar amendments increased pH, which was positively correlated with $^{226}$Ra and $^{137}$Cs activity concentrations. Therefore, it can be implied that pH has a great influence on radionuclides availability, a conclusion reported in the literature [22]. Additionally, it has been reported that media chemical parameters such as pH contributes to the various effect of radium mobility or adsorption [77,78], and we observed that $^{226}$Ra concentrations in samples varied with pH. The very significant correlations of the anthropogenic radionuclide $^{137}$Cs with C composition of the samples can be attributed to $^{137}$Cs fixation by organic C, in agreement with what has been previously reported [79]. The association of $^{40}$K with soil mineral fraction contributed to the lack of significant correlation of this radionuclide with samples' C composition [79].

The relative abundance of *Proteobacteria* (41.9–47.2%), *Bacteroidetes* (25.1–31.8%), *Acidobacteria* (5.2–8.3%) and *Actinobacteria* (2.6–4.5%) in the biochar-amended growth media reported in this study reflects their frequent presence in radionuclide contaminated environments [39,80–82]. For example, *Bryobacter* identified in our samples is known to withstand extreme environments including uranium-contaminated samples [83,84]. However, the above predominant bacteria phyla and genera did not have a significant correlation with the detected radionuclides in the present study.

At the phyla level, *Chloroflexi* and *Cyanobacteria* were the two phyla that significantly correlated with some of the detected radionuclides. The presence of *Chloroflexi* has been reported in natural uranium ores [16,85] and uranium-contaminated sites [16,86]. Elsewhere, *Chloroflexi* has been proposed to have the ability to degrade plant polymers, lignocellulosic material, and tolerate uranium and its associated radioactive progenies [86]. However, in our study, *Chloroflexi* was not correlated with any of the uranium progenies detected in this study but was correlated with $^{137}$Cs. To our knowledge, this study is the first to report a significant negative correlation between *Cyanobacteria* and radionuclides $^{226}$Ra, $^{40}$K, and $^{137}$Cs. But with respect to heavy metals, *Cyanobacteria* can produce polyphosphate granules for Cu immobilization that could be adsorbed to transport heavy metals into cells of *Cyanobacteria* [87,88].

The identified bacteria genera, *Cytophaga* and *Mucilaginibacter* significantly correlated with the detected radionuclides in the present study. These two bacteria genera, *Cytophaga* and *Mucilaginibacter* can be suggested to resist $^{226}$Ra, $^{40}$K, and $^{137}$Cs, and $^{226}$Ra and $^{137}$Cs contaminants, respectively, due to their negative relationships. This is in line with previous reports where *Mucilaginibacter* was identified as a metal-resistant bacterium [87].

## 5. Conclusions and Future Research

This study investigated the relationships among the physiochemical properties, carbon composition, R400, and the indigenous microbial communities and radionuclide concentrations in biochar-amended growth media. We concluded that the samples' pH significantly influences carbon composition, microbial composition, and the activity concentrations of the radionuclides. Simultaneously, *Acidobacteria*, *Chloroflexi*, and *Cyanobacteria* significantly correlated with carboxyl-C, phenolic-C, and aromatic-C. There were significantly negative relationships of *Cyanobacteria* with $^{226}$Ra, $^{40}$K, and $^{137}$Cs, which indicated resistance between *Cyanobacteria* and the radionuclides. *Chloroflexi* had a significant positive relationship

with $^{137}$Cs, suggesting tolerance. Like the phylum *Cyanobacteria*, the genera *Cytophaga* and *Mucilaginibacter* had significant negative relationships with $^{226}$Ra, $^{40}$K, and $^{137}$Cs, and $^{226}$Ra and $^{137}$Cs.

The relationships among C composition, microbial diversity, and radionuclide distributions from biochar-amended growth media might provide a better understanding of the physiochemical properties of such media and the development of plant growth-promoting growth media in the future. Further analyses are required for deeper knowledge of other types of soilless media to ascertain their properties and behavior with respect to radionuclides.

**Supplementary Materials:** The following supporting information can be downloaded at: https://www.mdpi.com/article/10.3390/applmicrobiol2030051/s1, Figure S1: Relative abundance of dominant bacteria phyla as detected in different % biochar amended soilless growth media; Figure S2: Carbon composition relationships with bacteria phyla. *a* and *b* represent correlation between aromatic and phenolic-C, and *Acidobacteria, c,* and *d* is the relationships between aromatic-C and total carbon (TC) and *Chloroflexi*, while e shows the relationship between phenolic-C and *Cyanobacteria*; Figure S3 Carbon composition relationships with bacteria phyla. *Cytophaga* with (***a***) carboxyl-C, (***b***) phenolic-C, and (***c***) aromatic-C. and *Mucilaginibacter* with (***a***) carboxyl-C, (***b***) phenolic-C, and (***c***) aromatic-C; Figure S4: (***a***) Aromatic-C relationship with *Terrimonas* and (***b***) phenolic-C relationship with *Acidibacter*; Figure S5: Relationships of C composition with radionuclides $^{226}$Ra and $^{137}$Cs; Figure S6: Relationship between total C and (***a***) $^{226}$Ra and (***b***) $^{137}$Cs; Figure S7: R400 relationships with bacteria phyla: (***a***) *Chloroflexi* and (***b***) *Patescibacteria*, bacteria genera: (***c***) *Terrimonas* and (***d***) *Cellvibrio*, and radionuclides: (***e***) $^{226}$Ra and (***f***) $^{137}$Cs.

**Author Contributions:** Conceptualization, G.K.O., M.A., L.N. and A.B..; methodology, G.K.O., M.A., L.N. and A.C.; validation G.K.O., A.C. and C.J.; formal analysis, G.K.O.; writing—original draft preparation, G.K.O.; writing—review and editing, G.K.O., M.A., L.N., A.C., A.B., A.P. and C.J. All authors have read and agreed to the published version of the manuscript.

**Funding:** Funding numbers 1901377 and 2200615, as well as USDA award number, respectively. Financial support from FAMU's title III program and the School of graduate studies is also acknowledged.

**Data Availability Statement:** Metagenomics data obtained from this study is available via National Center for Biotechnology Information's (NCBI's) Sequence Read Archive Bioproject accession # PRJNA773140.

**Acknowledgments:** Basic processing of the raw metagenomic data was performed by the University of Illinois at Chicago. The authors acknowledge faculty, staff, and students of the Nuclear Instrumentation Laboratory of Alcorn State University, Lorman, Mississippi for samples gamma spectroscopy analysis. We acknowledge the staff at FAMU-REC for growth media preparation. We also want to send our gratitude to Meenakshi Agarwal for the DNA extraction supervision and Djanan Nemours for MESTA analyses. Part of this study was made possible by support from the Title III Grant at Florida A&M University. A portion of this work was performed at the National High Magnetic Field Laboratory, which is supported by National Science Foundation Cooperative Agreement No. DMR-1644779* and the State of Florida.

**Conflicts of Interest:** The authors declare that they have no known competing financial interests or personal relationships that could have appeared to influence the work reported in this paper.

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
