# Peer review of "Characterization of Microbial Communities and Naturally Occurring Radionuclides in Soilless Growth Media Amended with Different Concentrations of Biochar"

_2673-8007, doi:10.3390/applmicrobiol2030051_

Round 1

Reviewer 1 Report

Applied Microbiology (ISSN 2673-8007)

Manuscript ID: applmicrobiol-1811852

Manuscript Title: "Interactions between Microbial Communities and Naturally Occurring Radionuclides in Soilless Growth Media Amended with Different Concentrations of Biochar"

Reviewer comments:

This current study has significant value by knowing the effects of Biochar derived from the pyrolysis of waste biomass in the absence of or little oxygen, on biotic and abiotic factors and biological properties of the soil. Moreover, soilless substrate as plant growth media has received great attention worldwide in recent years, and the study is important because little is known about biochar modified for microbial diversity of soilless growth media and levels of radioactivity. In addition, the topic fits well with the scope of Applied Microbiology-MDPI and the results are of interest to the scientific community. This study is well-designed and the methods are satisfy. However, the manuscript needs a major revision before publication. In addition, the innovation is sufficient and some of the discussion is inadequate. So, it will be deserved a major revision before consideration for publication in Applied Microbiology-MDPI.

Abstract and keywords:

  • Provide one introductory sentence in the summary about the problem of the study and then the purpose of conducting it
  • Write specific treatment name, which one is, produced better effects (Authors should write the best treatments that gave the highest results as a recommendation in the penultimate paragraph in the abstract section.)
  • Chang the keywords and write keywords that are not in the title of the manuscript (It is preferable not to write abbreviations in keywords, especially since these abbreviations are written for the first time. In addition, I suggest rephrasing some words because keywords should not repeat words from the title).
  • The abbreviation should be used after the full term. Please be consistent with the usage of all abbreviations. Pls revise the abbreviations in the whole part of MS.
  • At the end of the abstract, the authors have to write the general conclusion of this research.
  • Lines: 12-13:” A soilless substrate as plant growth media has been attracting high attention globally over the past five and half decades”. This sentence constitutes a broad meaning, please be specific by including specialized words that fit the topic of the manuscript.
  • Authors should write the best treatments that gave the highest results as a recommendation in the penultimate paragraph in the abstract section.

Introduction:

  • Lines 65 and 66: “The three major primordial radionuclides components source are from the decay series of 238U and 232Th, and 40K 66” what do you mean by 238U and 232Th, and 40K 66.
  • Lines 69-71: “For the past seven and half decades, the main sources of global anthropogenic radioactive contaminants in the environment have been atmospheric nuclear weapons tests, released from nuclear accidents such as the Chernobyl and Fukushima accidents, and mining [19 - 7121]. Pls rephrase and shorten.
  • Lines 76-79: “Radionuclides contaminated growth media for agriculture and horticulture crop production can potentially be a long-term source of radiation exposure to humans and the environment due to their accumulation by plants which in turn inhibit the growth and nutrition of various agricultural crops [25]” Pls rephrase and shorten.
  • Lines 84 - 86: “With all the 84extensive studies conducted with respect to bacteria interactions with contaminants in the agroecosystem, traditional soil has been used as the main growth media”. Pls add relevant ref.
  • The plagiarism should be reduced according to the journal's requirements.
  • Pls add a short paragraph about the importance and novelty of the study compared to previous studies in this regard.

Materials and methods:

  • The materials and methods section is well written but relevant references have to add to support them.
  • It is important to add relevant and recent references regarding all methods used in this section.
  • Pls shorten the materials and methods section; there are more unnecessary details in several parts.

Results and discussion

  • Figure 3. The resolution is not good.
  • Please highlight more specifically the objective of the work.
  • The results part is very long, this part should be shortened as much as possible in order to reach the intended meaning directly for all the studied characteristics
  • In the discussion section, conjunctions should be used to show the relationship between sentences.
  • Lines 425-427:” What do you mean by “In agreement with the report, the present study showed the abundance of Flavobacterium in the growth media samples collected, and its negative correlation with R400 indicates organic matter mineralization.
  • The resolution of some figures was not clear enough, so pls pay attention to this comment.
  • Please, make an effort to synthesize the text avoiding redundancies and repetitions in the discussion.
  • Discussion in several parts is confusing, suggesting rewriting.
  • Some parts of the discussion sentences need clarification and interpretation, and recent references need to be used as much as possible.
  • The discussion should be better organized. It is important to try to better deepen and explain.

Conclusions: the authors should write a summary of the current work in short sentences so that I, as a reader of this article, can understand what the article ended up being.

References

  • The references part must be written according to the requirements of the journal, where the year of publication is put in its correct place
  • The number of references is about 94 ref. I think it is satisfied. 11 of them during the last five years. So, pls delete the old ones and avoid repetition. There is a recent ref. (2020-2022) in the same trend of the topic of this MS, pls pay attention to this point and cross-check all the references for mistakes, and follow the journal style of reference input.

General comments:

  • To summarize, this is a good study, which certainly merits publication after a major revision.
  • The manuscript contains some typo errors; please revise it very carefully. A careful revision of the English Grammar is required. So, language needs to be improved thoroughly

Reviewer 2 Report

I reviewed the article with title: Interactions Between Microbial Communities and Naturally Occurring Radionuclides in Soilless Growth Media Amended with Different Concentrations of BiocharThe article topic is intriguing and promising in the area. Overall, the article structure and content are suitable. I am pleased to send you moderate level comments, which need to be corrected before publication. Please consider these suggestions as listed below.

1.      The title seems good, but the abstract seems to be wired. Please add one more introductory line of your objective in beginning of abstract.

2.      Research gap should be delivered on more clear way with directed necessity for the future research work.

3.      Introduction section must be written on more quality way, i.e., more up-to-date references addressed. Please target the specific gap.

4.      The novelty of the work must be clearly addressed and discussed, compare previous research with existing research findings and highlight novelty.  

5.      What is the main challenge? Please highlight in the introduction part.

6.      Please check the abbreviations of words throughout the article. All should be consistent.

7.      Please include all chemical/instrumentation brand name and other important specification.

8.      Please add chemical reagents section and stated all chemical with brand specifications.

9.      Page 1 Line 42 need this reference with existing reference 5. Please cite- Toxicology and environmental application of carbon nanocomposite.

10.   The main objective of the work must be written on the more clear and more concise way at the end of introduction section.

11.   Please provide space between number and units. Please revise your paper accordingly since some issue occurs on several spots in the paper. 

12.   Regarding the replications, authors confirmed that replications of experiment were carried out. However, these results are not shown in the manuscript, how many replicated were carried out by experiment?

13.   Please provide high quality image for figure 3.

14.   Section 5 should be renamed by Conclusion and Future perspectives. Conclusion section is missing some perspective related to the future research work, quantify main research findings, highlight relevance of the work with respect to the field aspect.

15.   To avoid grammar and linguistic mistakes, minor level English language should be thoroughly checked. Please revise your paper accordingly since several language issue occurs on several spots in the paper.

16.   Reference formatting need carefully revision. All must be consistent in one formate. Please follow the journal guidelines.

Reviewer 3 Report

In this manuscript, Osei et al studied biochar soilless media microbial composition and radioactivity using different techniques, I have some comments as described in below:

Major comments:

The authors in Table 2 showed that the activity concentration of 232Th was 4.98 units in a 3% biochar amended growth media, which is 5 times higher than the other values of 232Th. Perhaps there is a need to explain this significant data discrepancy?

“This scenario is contrary to what was observed in this study where Cytophaga has a positive relationship with R400 indicating preservation of organic matter.” It’s a really curious contradiction. How do the authors explain it, or are there no assumptions at this stage of research? Is it possible that this is correlated with negative relationship with pH in this study?

Minor comments

Line 40 add dot at the end of a sentence “…such as biochar”

Line 149 “using the Equation…” change to “Using the Equation…”

Line 153 “DNA Extraction, Quantification, Purity, and metagenomics” perhaps it's better change to “DNA Extraction, Quantification and Purity. Metagenomics”

Line 209 “2.3.6.235. U, 226Ra, 232Th, 40K and 137Cs” change to “2.3.6. 235U, 226Ra, 232Th, 40K and 137Cs”, move index for uranium to uppercase

Line 222 “National Center for Biotechnology Information’s (NCBI's) Sequence Read Archive Bioproject accession # PRJNA773140 can be used to access the amplicon metagenomic 16S sequences…” may be a mistake in the Bioproject number, as it is not found when accessing the NCBI website

Line 287 “… FAMU-CE growth. media." maybe it means “… FAMU-REC growth media.”

Line 324 “…(Figure SI5a, b and c)…” maybe it means “…(Figure S5a, b and c)…”

Line 362 “…majority of enzymes and proteins become denatured and inactivated under high acidic...” change to “…Majority of enzymes and proteins become denatured and inactivated under high acidic...”

Lines 486-488 change the font for phylum and genera to Italics

Round 2

Reviewer 1 Report

Applied Microbiology (ISSN 2673-8007)

Manuscript ID: applmicrobiol-1811852

Manuscript Title: "Interactions between Microbial Communities and Naturally Occurring Radionuclides in Soilless Growth Media Amended with Different Concentrations of Biochar"

Reviewer comments:

After reviewing the entire manuscript and reviewing my previous comments on the manuscript, I found that the manuscript has improved now, but the following must be taken into account:

Abstract and keywords:

·        Chang the keywords and write keywords that are not in the title of the manuscript (It is preferable not to write abbreviations in keywords, especially since these abbreviations are written for the first time. In addition, I suggest rephrasing some words because keywords should not repeat words from the title).

·        The abbreviation (R400) should be used after the full term. Please be consistent with the usage of all abbreviations. Pls revise the abbreviations in the whole part of MS.

Introduction:

·        Pls add a short paragraph about the importance and novelty of the study compared to previous studies in this regard.

Materials and methods:

·        It is important to add relevant and recent references regarding all methods used in this section.

Results and discussion

·        Figure 3. The resolution is still not good.

·        Please highlight more specifically the objective of the work.

·        In the discussion section, conjunctions should be used to show the relationship between sentences.

·        Please, make an effort to synthesize the text avoiding redundancies and repetitions in the discussion.

·        The discussion should be better organized. It is important to try to better deepen and explain.

References

·        The references part must be written according to the requirements of the journal, where the year of publication is put in its correct place. Pls carefully revise

·        Add recent ref. (2020-2022) if possible and cross-check all the references for mistakes, and follow the journal style of reference input.

·        The manuscript contains some typo errors; please revise it very carefully. A careful revision of the English Grammar is required. So, language needs to be improved thoroughly

Reviewer 2 Report

The authors have addressed recommended changes. I recommend it for publication in your journal
